# Bacterial Resistance to Antibiotics and Clonal Spread in COVID-19-Positive Patients on a Tertiary Hospital Intensive Care Unit, Czech Republic

**DOI:** 10.3390/antibiotics11060783

**Published:** 2022-06-08

**Authors:** Lenka Doubravská, Miroslava Htoutou Sedláková, Kateřina Fišerová, Vendula Pudová, Karel Urbánek, Jana Petrželová, Magdalena Röderová, Kateřina Langová, Kristýna Mezerová, Pavla Kučová, Karel Axmann, Milan Kolář

**Affiliations:** 1Department of Anesthesiology, Resuscitation and Intensive Care, University Hospital Olomouc, I. P. Pavlova 6, 779 00 Olomouc, Czech Republic; lenka.doubravska@fnol.cz (L.D.); karel.axmann@fnol.cz (K.A.); 2Department of Microbiology, University Hospital Olomouc, I. P. Pavlova 6, 779 00 Olomouc, Czech Republic; katerina.fiserova@fnol.cz (K.F.); jana.petrzelova@fnol.cz (J.P.); pavla.kucova@fnol.cz (P.K.); 3Department of Microbiology, Faculty of Medicine and Dentistry, Palacky University Olomouc, Hnevotinska 3, 779 00 Olomouc, Czech Republic; vendula.pudova@upol.cz (V.P.); magdalena.roderova@upol.cz (M.R.); kristyna.mezerova@upol.cz (K.M.); milan.kolar@fnol.cz (M.K.); 4Department of Pharmacology, Faculty of Medicine and Dentistry, Palacky University Olomouc, Hnevotinska 3, 779 00 Olomouc, Czech Republic; karel.urbanek@fnol.cz; 5Department of Medical Biophysics, Faculty of Medicine and Dentistry, Palacky University Olomouc, Hnevotinska 3, 779 00 Olomouc, Czech Republic; langova@tunw.upol.cz

**Keywords:** COVID-19, bacteria, antibiotics, multidrug resistance

## Abstract

This observational retrospective study aimed to analyze whether/how the spectrum of bacterial pathogens and their resistance to antibiotics changed during the worst part of the COVID-19 pandemic (1 November 2020 to 30 April 2021) among intensive care patients in University Hospital Olomouc, Czech Republic, as compared with the pre-pandemic period (1 November 2018 to 30 April 2019). A total of 789 clinically important bacterial isolates from 189 patients were cultured during the pre-COVID-19 period. The most frequent etiologic agents causing nosocomial infections were strains of *Klebsiella pneumoniae* (17%), *Pseudomonas aeruginosa* (11%), *Escherichia coli* (10%), coagulase-negative staphylococci (9%), *Burkholderia multivorans* (8%), *Enterococcus faecium* (6%), *Enterococcus faecalis* (5%), *Proteus mirabilis* (5%) and *Staphylococcus aureus* (5%). Over the comparable COVID-19 period, a total of 1500 bacterial isolates from 372 SARS-CoV-2-positive patients were assessed. While the percentage of etiological agents causing nosocomial infections increased in *Enterococcus faecium* (from 6% to 19%, *p* < 0.0001), *Klebsiella variicola* (from 1% to 6%, *p* = 0.0004) and *Serratia marcescens* (from 1% to 8%, *p* < 0.0001), there were significant decreases in *Escherichia coli* (from 10% to 3%, *p* < 0.0001), *Proteus mirabilis* (from 5% to 2%, *p* = 0.004) and *Staphylococcus aureus* (from 5% to 2%, *p* = 0.004). The study demonstrated that the changes in bacterial resistance to antibiotics are ambiguous. An increase in the frequency of ESBL-positive strains of some species (*Serratia marcescens* and *Enterobacter cloacae*) was confirmed; on the other hand, resistance decreased (*Escherichia coli*, *Acinetobacter baumannii*) or the proportion of resistant strains remained unchanged over both periods (*Klebsiella pneumoniae*, *Enterococcus faecium*). Changes in pathogen distribution and resistance were caused partly due to antibiotic selection pressure (cefotaxime consumption increased significantly in the COVID-19 period), but mainly due to clonal spread of identical bacterial isolates from patient to patient, which was confirmed by the pulse field gel electrophoresis methodology. In addition to the above shown results, the importance of infection prevention and control in healthcare facilities is discussed, not only for dealing with SARS-CoV-2 but also for limiting the spread of bacteria.

## 1. Introduction

Coronavirus disease 2019 (COVID-19) is a viral disease caused by severe acute respiratory syndrome coronavirus 2 (SARS-CoV-2), which was first identified in China in December 2019 and has caused an ongoing pandemic. As of 30 May 2022, more than 523 million cases and over 6 million deaths have been reported globally [1]. Due to the current COVID-19 pandemic, there have been changes to the organization of healthcare, hygiene and epidemiology, mobility, human behavior and lifestyle throughout the world. The main priority has been controlling the COVID-19 pandemic, preventing/reducing spread of this infection and decreasing the mortality rates. As a result, less attention has been paid to another pandemic, one that is longer and more insidious, but no less dangerous and current, infectious diseases caused by multidrug-resistant (MDR) bacteria. According to the well-known O’Neill’s estimate of the increase in bacterial resistance, diseases caused by MDR bacteria may cause as many as 10 million deaths per year by 2050 [2].

Because of the current COVID-19 pandemic, trends in the development of antibiotic resistance are very likely to change. This will be contributed to by many factors, causing resistance to either rise or decline. Such factors may include higher numbers of patients with serious conditions and altered immune system function, overwhelmed healthcare systems, overworked health professionals, implementation of special hygiene and sanitary measures, increased use of disinfectants, non-urgent care and surgery postponement, limited antimicrobial stewardship due to changing focus to COVID-19 and increased use of antibiotics for bacterial coinfections or superinfections.

Bacterial coinfections and secondary bacterial infections have been and always will be important for morbidity and mortality of patients with viral infections [3,4,5,6,7]. For example, the 2009 H1N1 flu pandemic is estimated to have caused approximately 300,000 deaths, with 30–55% of complications associated with secondary bacterial infection [8]. According to recent data, 3–7% of patients with COVID-19 staying in general wards have community-acquired bacterial coinfections [9,10,11,12,13,14]. In ICUs, the proportion is reported to be higher, ranging from 14% to 28% [15,16]. Therefore, community-acquired coinfections are rare in COVID-19 patients, whereas the likelihood of bacterial superinfection increases with longer hospital stays and as a result of other factors (respiratory insufficiency, taking vasopressors, staying in hospital for more than seven days, with the introduction of invasive devices, etc.).

Rapid identification of bacterial coinfection is of the utmost importance and so is antibiotic therapy in concordance with the principles of rational antibiotic policy, not only for reducing the morbidity and mortality of COVID-19 patients but also as a key tool for the implementation of antimicrobial stewardship during the pandemic. Data show that approximately 70% of COVID-19 patients have been treated with antibiotics, mostly broad-spectrum and empirical ones [9,10]. It is assumed that increased consumption of antibiotics for bacterial coinfections and superinfections in COVID-19 may result in higher selection pressure on bacterial pathogens and rising antibiotic resistance.

Of course, all these factors are debatable and it has been hypothesized that the changes may have an impact on bacterial resistance to antibiotics. To determine whether the bacterial spectrum of pathogens and their antibiotic resistance changed in patients with critical COVID-19 in comparison to the pre-COVID-19 period, there was a close cooperation between the Department of Microbiology and the Department of Anesthesiology, Resuscitation and Intensive Care, University Hospital Olomouc, Czech Republic.

The University Hospital Olomouc is one of the largest healthcare facilities in the Czech Republic (1200 beds), providing medical care to approximately 925,000 outpatients and 50,000 inpatients per year. In the hospital, the highest level of intensive care is provided by the Department of Anesthesiology, Resuscitation and Intensive Care. The department admits critically ill individuals with any diagnoses, the only exception being cardiac surgery patients. Admissions are decided based on the severity of patients’ conditions, with more than 95% being intubated and on mechanical ventilation prior to or shortly after their arrival to the department. There are ten beds, of which four are in separate cubicles and six are in an open space. Common hygiene measures are in place to prevent healthcare-associated infections from developing. Drugs and infusions are prepared in a special area under sterile conditions. Disposable gloves are used for all activities. Surfaces are regularly disinfected. The directions for disinfectant use are issued by the hospital’s Department of Hygiene while considering the local epidemiological situation. Medical devices and tools are reserved for each patient and are not shareable. All invasive procedures are performed by staff members wearing personal protective equipment under strict sterile conditions (changing wound dressings, caring for invasive entry sites). Ventilators are equipped with single-use circuits and filters. In mechanically ventilated patients, closed suctioning systems are used. All these components are regularly replaced and the exhaled air is conducted to the central system to prevent contamination of the environment from patients’ airways.

In the spring of 2020, the Department of Anesthesiology, Resuscitation and Intensive Care was transformed into an intensive care unit (ICU) for COVID-19 patients. During the main waves of the pandemic, between the autumn of 2020 and the spring of 2021, the capacity of the department was extended to as many as 35 beds (350%), mostly (approximately 70%) in the open-space arrangement, divided into five halls. The situation was unique in that all patients were admitted for SARS-CoV-2 infection. In 97% of cases, patients suffered from acute respiratory distress syndrome (ARDS), sepsis or septic shock, that is, critical COVID-19, as classified by the World Health Organization (WHO) [17].

Only a very small minority of patients were admitted to the department for other serious conditions requiring intensive care, such as trauma and stroke, or postoperative intensive care. Although they did not suffer from COVID-19 pneumonia and respiratory distress, their admission to the department was warranted to ensure their isolation following a positive PCR test for SARS-CoV-2.

Upon their admission to the department, all critical COVID-19 patients were started on oxygen therapy: either High-Flow Nasal Oxygen Therapy (HFNOT), as a method of choice in patients with mild ARDS, as defined by the Berlin criteria [18,19], or mechanical ventilation, or extracorporeal membrane oxygenation (ECMO), where applicable.

The objective of the study was to show whether the spectrum of bacterial pathogens and their resistance to antibiotics changed during the worst part of the COVID-19 pandemic among the patients in the Department of Anesthesiology, Resuscitation and Intensive Care, University Hospital Olomouc, in comparison to the pre-COVID-19 period. Additionally, the present study aimed to establish whether the potential changes in the distribution of pathogens and their resistance were caused by the selective pressure of antibiotics or by clonal spread of identical bacterial isolates from patient to patient.

## 2. Materials and Methods

The observational retrospective study comprised patients staying in the Department of Anesthesiology, Resuscitation and Intensive Care over two six-month periods: 1 November 2018 to 30 April 2019 (prior to the COVID-19 pandemic) and 1 November 2020 to 30 April 2021 (time of the COVID-19 pandemic). Informed consent was not required from patients since all participants underwent standard diagnostic and therapeutic interventions; the institutional ethics committee approved the realization of the study. Both groups of patients before and during the COVID-19 pandemic were compared in qualitative characteristics with chi-squared test and analyzed with Mann–Whitney U-test for age. The data were processed with IBM SPSS Statistics for Windows, Version 23.0 (IBM Corp., Armonk, NY, USA).

Initial therapy for hospital-acquired pneumonia (HAP) in mechanically ventilated patients routinely treated before the COVID-19 pandemic was based on the administration of the broad-spectrum beta-lactam antibiotics amoxicillin/clavulanic acid or ampicillin/sulbactam (in patients with early HAP), or ceftazidime or piperacillin/tazobactam or meropenem (in patients with late HAP) in combination with gentamicin or amikacin (in patients with sepsis). If MRSA was suspected, linezolid or vancomycin was added to the combination. If severe community-acquired pneumonia (CAP) requiring hospitalization was diagnosed, co-aminopenicillin or cefotaxime in combination with clarithromycin was indicated.

During the pandemic, all patients with critical COVID-19 received initial antibiotic therapy, namely cefotaxime combined with clarithromycin. After bacterial coinfection was ruled out or inflammatory biochemical parameters (C-reactive protein, procalcitonin, leukocytes) decreased and airway secretion microbiology tests yielded negative results, antibiotic therapy was discontinued after three to five days. Conversely, following an increase in inflammatory parameters and/or positive airway secretion tests or development of bacterial-ventilator-associated pneumonia, initial antibiotic therapy was switched to targeted antibiotic therapy based on isolation of bacterial pathogens and determination of their susceptibility/resistance to antibacterial agents.

In patients hospitalized between 1 November 2020 and 30 April 2021, COVID-19 was diagnosed by direct virus detection with RT-PCR identifying three specific gene areas of viral RNA in nasopharyngeal and/or oropharyngeal swabs [20].

Over both periods, clinical samples from the upper or lower airways (nasopharyngeal swabs or sputum in non-intubated patients, airway secretions in intubated patients), urine and, if sepsis was suspected, blood cultures were regularly collected (on admission and then twice weekly) in all patients as a screening of microbial colonization. In case of clinical suspection for infectious complications, samples were also taken from particular sites (punctates, pus, swabs from wounds, vascular cannulas, secretions from drains, etc.). Bacterial pathogens were identified using standard microbiology techniques with an MALDI-TOF MS system (Biotyper Microflex, Bruker Daltonics). For each patient, a single strain of each species with particular antibiogram, isolated as the first one from a particular clinical sample, was included in the study. In clinically important isolates susceptibility to antibiotics was determined with a standard microdilution method in accordance with the EUCAST criteria [21]. To ensure quality control, the following reference bacterial strains were used: *Escherichia coli* ATCC 25922, *Pseudomonas aeruginosa* ATCC 27853, *Staphylococcus aureus* ATCC 29213 and *Enterococcus faecalis* ATCC 29212. Production of broad-spectrum ESBL and AmpC beta-lactamases and carbapenemases was detected by phenotypic tests [22,23] and confirmed by PCR detection of the relevant genes. All *Staphylococcus aureus* strains were tested for resistance to methicillin using selective diagnostic chromogenic media (Colorex/TM/MRSA, TRIOS) and an immunochromatographic assay for the detection of PBP2a (PBP2a SA Culture Colony Test, Alere^TM^). Positive results were confirmed by detection of the *mecA* gene [24]. Resistance of vancomycin-resistance enterococci was confirmed by detection of the *vanA* and *vanB* genes [25].

The clonality of *Klebsiella pneumoniae* strains (12 isolates stored in the pre-COVID-19 period) was assessed with pulsed-field gel electrophoresis (PFGE). During the pandemic, clonality was determined for isolated strains of *Klebsiella pneumoniae* (41 strains)*, Serratia marcescens* (7 strains)*, Acinetobacter baumannii* (7 strains) and *Burkholderia multivorans* (25 strains). Bacterial DNA was isolated from cells grown on blood agar (TRIOS, Czech Republic) for 18 h. Bacterial suspension was prepared with cell suspension buffer according to PulseNet PFGE protocol [26]. Agarose blocks were made by modified PFGE protocol with SDS-based lysis buffer [27]. DNA was cleaved with the enzyme XbaI (Takara Biotechnology, Kyoto, Japan) and then separated by PFGE in 1.2% agarose gel under the following conditions: 24 h, 6 V·cm^−1^, initial switch time 2 s, final switch time 35 s. The resulting restriction profiles were compared with the GelCompar II software (Applied Maths, Kortrijk, Belgium). Restriction profiles reaching 95% similarity were considered identical. *Burkholderia multivorans* isolates were compared using random amplification of polymorphic DNA (RAPD) [28].

Over both periods, antibiotic consumption was assessed according to the 2020 ATC/DDD system and expressed as numbers of defined daily doses (DDD) for individual classes of antibiotics [29]. To assess how the distribution of consumed antibiotic has changed, the percentage was calculated as a proportion of the total consumed volume of all antibiotics.

## 3. Results

Over the six-month period between 1 November 2018 and 30 April 2019, a total of 189 patients stayed in the Department of Anesthesiology, Resuscitation and Intensive Care. The patients’ mean age was 63 years (median 65; range 18–97). There were 136 males (72%). Mechanical ventilation and high-flow oxygen therapy (HFNOT) were delivered to 97% and 5% of patients, respectively. Two patients (1%) received noninvasive ventilation. Early admissions (within 48 h of arrival to the hospital) accounted for 43% of cases; the remaining 57% of patients were transferred to the department from other departments or healthcare facilities after more than 48 h of stay. The in-hospital mortality rate was 39%. The distribution of patients by diagnosis on admission in the pre-COVID-19 period is illustrated in Figure 1. The group of patients is characterized in Table 1.

A total of 789 clinically important bacterial isolates were cultured during the pre-COVID-19 period. Numbers of clinically relevant bacterial isolates from particular biological materials are shown in Table 2. The most frequent etiologic agents causing nosocomial infections, in particular HAP, urinary tract infections and sepsis, i.e., isolated from blood stream, lower respiratory tract and urine, were strains of *Klebsiella pneumoniae* (17%), *Pseudomonas aeruginosa* (11%), *Escherichia coli* (10%)*,* coagulase-negative staphylococci (9%), *Burkholderia multivorans* (8%), *Enterococcus faecium* (6%), *Enterococcus faecalis* (5%), *Proteus mirabilis* (5%) and *Staphylococcus aureus* (5%).

Over the comparable COVID-19 period (1 November 2020 to 30 April 2021), a total of 372 patients stayed in the department. During the pandemic, only SARS-CoV-2-positive patients were admitted to the department. The mean age was 64 years (median 67; range 22–90). Mechanical ventilation and HFNOT were received by 67% and 76% of patients, respectively, with nearly half of patients (49%) receiving both HFNOT and mechanical ventilation. Fifteen patients were on noninvasive ventilation, but for only a few hours. Early admissions accounted for 50% of cases. The pre-discharge mortality rate was 43%. The group of patients is characterized in Table 1. No significant differences in gender, age, timing of admission or mortality were observed between either patient group (Table 1).

As many as 1500 bacterial strains were isolated during the COVID-19 period. Numbers of clinically relevant bacterial isolates from particular biological materials are shown in Table 2. The order of the most common etiologic agents causing nosocomial infections, in particular HAP, urinary tract infections and sepsis was slightly different in comparison to the pre-COVID-19 period: *Enterococcus faecium* (19%), *Klebsiella pneumoniae* (18%), coagulase-negative staphylococci (10%), *Pseudomonas aeruginosa* (9%), *Serratia marcescens* (8%), *Burkholderia multivorans* (7%) and *Klebsiella variicola* (6%). Changes in the percentages of bacterial species from the total number of etiological agents causing HAP, urinary tract infections and sepsis during the pandemic, as compared with the similar pre-COVID-19 period, are shown in Figure 2. While considerable increases are seen in *Enterococcus faecium* (from 6% to 19%, *p* < 0.0001), *Klebsiella variicola* (from 1% to 6%, *p* = 0.0004) and *Serratia marcescens* (from 1% to 8%, *p* < 0.0001), there were significant decreases in *Escherichia coli* (from 10% to 3%, *p* < 0.0001), *Proteus mirabilis* (from 5% to 2%, *p* = 0.004) and *Staphylococcus aureus* (from 5% to 2%, *p* = 0.004).

Table 3 documents the resistance of bacterial pathogens to antibiotics before and during the COVID-19 pandemic. The results show that some species considerably increased their resistance during the pandemic. *Serratia marcescens* increased its resistance to third- and fourth-generation cephalosporins (cefotaxime, ceftazidime, cefepime) from 14%, 0%, respectively, to 70% as well as to ciprofloxacin (from 0% to 67%), gentamicin (from 0% to 69%), tigecycline (from 14% to 100%) and co-trimoxazole (from 0% to 71%). A significant increase in resistance to third-generation cephalosporins and piperacillin/tazobactam was also noted in *Enterobacter cloacae* (from 33% to 68%). *Stenotrophomonas maltophilia* increased its resistance significantly to ciprofloxacin (from 70% to 100%).

On the other hand, resistance of some bacterial species significantly decreased, for example, resistance to meropenem in the case of *Pseudomonas aeruginosa* (from 51% to 22%), *Klebsiella pneumoniae* (from 2% to 0%), *Burkholderia multivorans* (from 29% to 2%) or *Acinetobacter baumannii* (from 71% to 0 %). Strains of *Escherichia coli* decreased their resistance significantly to gentamicin (from 19% to 0%). In the case of *Acinetobacter baumannii*, there was a significant reduction in resistance to many other antibiotics, such as ampicillin/sulbactam, ceftazidime, cefepime, piperacillin/tazobactam, co-trimoxazole, gentamicin, amikacin and ciprofloxacin. Finally, *Enterococcus faecium* reduced its resistance significantly to tigecycline from 13% to 0%.

The consumption of antibacterial agents over the two periods is shown in Table 4. During the COVID-19 pandemic, a significant decrease in the consumption of amoxicillin/clavulanic acid was noted. On the other hand, consumption of cefotaxime was significantly higher than in the pre-COVID-19 period.

PFGE proved the clonal spread of bacterial strains during the pandemic. Among a total of 41 *Klebsiella pneumoniae* isolates obtained during the pandemic, 14 pulsotypes were identified. Only 7 (17%) isolates had unique restriction profiles; the remaining 34 strains were divided into seven groups based on their comparison. The largest cluster included 19 isolates with a coefficient of similarity (CS) of more than 95%; thus, the isolates may be considered identical. Additionally, there was one group of four identical isolates, one group of three identical isolates and four pairs of identical isolates (CS > 95–100%). Seven analyzed isolates of *Serratia marcescens* included three pairs of identical isolates (CS > 95–100%) and one unique strain. A group of three and one couple of identical isolates were detected among 7 *Acinetobacter baumannii*. Among 25 isolates of *Burkholderia multivorans*, 20 (80%) belonged to one RAPD type. The results clearly illustrate clonal spread during the COVID-19 pandemic, as seen from time axes for some of the pathogens (Figure 3 and Figure 4). Unlike the proven clonal spread during the COVID-19 pandemic, all tested *Klebsiella pneumoniae* strains that were available from the pre-COVID-19 period (12 isolates) had unique restriction profiles.

## 4. Discussion

The present study showed a decrease in the consumption of essential antibiotics, such as amoxicillin/clavulanic acid, cefuroxime and gentamicin, and a considerable increase in the use of broad-spectrum and reserve antibiotics (significantly shown in cefotaxime, but non-significantly also in amikacin, linezolid and colistin). This is because all patients staying in the Department of Anesthesiology, Resuscitation and Intensive Care at the worst time of the COVID-19 pandemic received initial broad-spectrum antibiotic therapy, namely cefotaxime combined with clarithromycin. This was warranted by their serious condition combined with troublesome differential diagnosis between viral and bacterial pneumonia. The commonly used parameters (cough, dyspnea, chest pain, auscultation and chest X-ray findings) are non-specific for bacterial inflammation. Common biochemical parameters, such as C-reactive protein, white blood cell count or ferritin, cannot be relied upon either as their levels are increased in COVID-19 patients. At the present time, the best biomarker appears to be procalcitonin [30,31]. Finally, a fact that played a role in the decision about the initial antibiotic therapy was that, while bacterial coinfection is relatively frequent (11–35%) in flu pneumonia [15,32], analogical data for COVID-19 were missing in the early pandemic phase.

The present study, however, failed to clearly confirm an overall rise in bacterial resistance during the COVID-19 pandemic as compared with the pre-pandemic period. Even though the frequency of MDR strains of some bacterial species increased (e.g., *Serratia marcescens, Enterobacter cloacae*), the resistance of other bacterial species to some antibiotics decreased (*Acinetobacter baumannii*, *Pseudomonas aeruginosa*, *Enterococcus faecium*, *Escherichia coli*) or remained unchanged (*Klebsiella pneumoniae*, vancomycin-resistant *Enterococcus faecium*). Data from the literature are also ambiguous, with increased/decreased resistance in some species to particular antibiotics or resistance remaining stable [33,34,35,36,37,38,39]. It may be due to the short interval of observation; probably, more time will be needed to state how the COVID-19 pandemic has affected the trends in bacterial resistance to antibiotics.

A surprising finding is the decreased resistance of *Pseudomonas aeruginosa* to meropenem. A study by Kolar et al. documented a rising frequency of *Pseudomonas aeruginosa* strains resistant to meropenem in association with its consumption [40]. Comparison of meropenem use in the two studied periods shows that although the absolute consumption increased, its proportion among antibiotics remained unchanged. That could be the reason why the selection pressure of meropenem was not expressed in the sense of rising antibiotic resistance. In comparison, both absolute and relative consumption of cefotaxime increased; the frequency of ESBL-positive strains increased in *Serratia marcescens* and *Enterobacter cloacae.*

Interestingly, the present study found that *Acinetobacter baumannii*, highly resistant to meropenem, aminoglycosides, ciprofloxacin and co-trimoxazole in the pre-COVID-19 period, decreased its resistance during the pandemic to zero (8% in ciprofloxacin, respectively). This may be explained by the “small number error” (only seven isolates detected before the pandemic) and by the clonal spread of a susceptible strain during the pandemic (a significant increase in number of strains during the pandemic period). In the COVID-19 period, clonal spread was found to be more significant in *Klebsiella pneumoniae* and *Burkholderia multivorans.* However, clonal spread of *Acinetobacter baumannii* and *Serratia marcescens* in smaller clusters was also proved.

Using the Pulse Gel Electrophoresis (PFGE) methodology, the spread of identical bacterial strains among COVID-19 patients in the Department of Anesthesiology, Resuscitation and Intensive Care was revealed. PFGE was considered to be a gold standard for bacterial typing for a long time and even now, in cases when it is redundant to use whole-genome sequencing (e.g., evaluation of potential clonal spread within local outbreak as in our case), this method is still suitable and useful [41]. Studies, which were performed in this department in the previous years (2011–2021) using PFGE, did not reveal significant clonal spread of bacterial pathogens [42,43,44]. The clonal spread was observed also in other countries, e.g., in Spain in *Klebsiella pneumoniae* strains, in the USA in *Escherichia coli* strains and in Mexico in *Acinetobacter baumannii* strains [45,46,47]. In the Department of Anesthesiology, Resuscitation and Intensive Care, the clonal spread of vancomycin-resistant enterococci was also demonstrated at the time of the pandemic [48].

The detection of clonal spread gives an opportunity to discuss its causes. In the first place, it may be associated with the use of HFNOT, as well as the fact that most of the beds were not placed in isolation cubicles.

Given the extensive use of HFNOT during the pandemic waves [19,49,50,51,52], many studies on the potential contribution of HFNOT to horizontal spread of infectious particles and the risk of staff members and other patients being infected have been published recently [53,54,55,56,57,58,59,60,61].

However, the use of HFNOT is not the only possible cause of the detected clonal spread. Other causes arise from the exceptional situation during the pandemic: the department was overloaded by a large number of patients and it was not possible to change the space arrangement of the halls, which would ensure the isolation of the patients (in each hall, more than five patients on HFNOT were staying in a single open space and the distance between them was less than two meters to increase the number of beds available). To cope with the large numbers of patients, some of the staff members were temporarily transferred from other hospital departments. This was associated with an increased number, turnover and movement of medical and non-medical staff. Patients were also transferred more frequently between hospital departments as their health status changed.

Other reasons for clonal spread may include lack of personal protective equipment at the beginning of the pandemic and unawareness, how to use them properly when manipulating patients (pronation, rehabilitation, etc.) Some of the strict hygiene measures may not have been adhered to by the staff in the overloaded department, and last, but not least, the possibility of contamination of the environment, such as infusions, bed-side equipment, disinfectant containers and ultrasound gels, may have also played a role, as described in the literature [62,63,64].

After the detection of the clonal spread (spring 2021), the following measures were implemented in cooperation with the hospital’s Department of Hygiene: one-time decontamination of the environment, one layer of barrier protection was added (plastic jacket) when manipulating a single patient, disinfection regimes were modified, re-education of the staff was performed, cleaning of the environment was intensified and compliance with hygienic–epidemiological measures was strictly controlled.

The authors are aware of the limitations of this work, namely, it is single-center, retrospective study, which compares shorter time periods (6 months). On the other hand, the methodology for the collection of biological materials, microbiological examinations and genetic analysis of clonal spread over both examined periods was carried out in the same way and did change, which could be emphasized as a strength of this study. In the next study, the frequency of bacterial co-infections and superinfection, etiology, antibiotic therapy and clinical outcome of COVID-positive patients should be described, which are objectives that were not included in this current study.

## 5. Conclusions

The present study showed ambiguous changes in the resistance of bacterial pathogens during the COVID-19 pandemic in comparison to the pre-COVID-19 period. There was an increase in the frequency of ESBL-positive strains of some species (*Serratia marcescens* and *Enterobacter cloacae*); on the other hand, resistance decreased (e.g., in *Acinetobacter baumannii* or in some Gram-negative bacteria to meropenem) or the proportion of resistant strains remained unchanged over both periods (*Klebsiella pneumoniae*, vancomycin-resistant *Enterococcus faecium*). In the Department of Anesthesiology, Resuscitation and Intensive Care, bacterial pathogens were observed to spread clonally at the worst time of the COVID-19 pandemic as compared with the pre-pandemic period. Given that fact, the adequacy of the existing hygiene and epidemiological measures and adherence were reassessed and corrected.

## Figures and Tables

**Figure 1 antibiotics-11-00783-f001:**
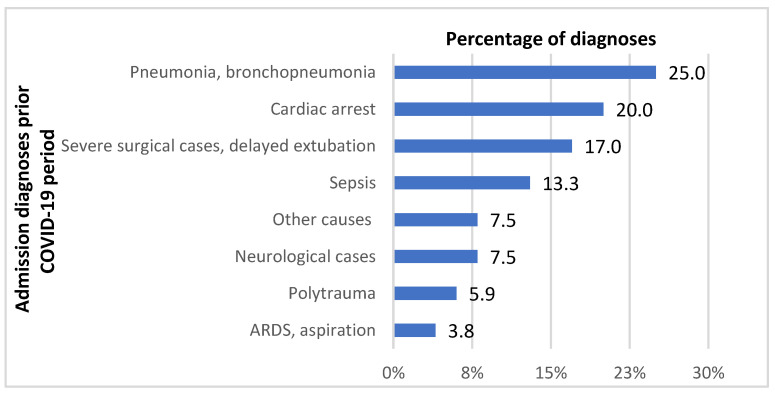
Distribution of admission diagnoses in the Department of Anesthesiology, Resuscitation and Intensive Care before the COVID-19 pandemic. ARDS—acute respiratory distress syndrome.

**Figure 2 antibiotics-11-00783-f002:**
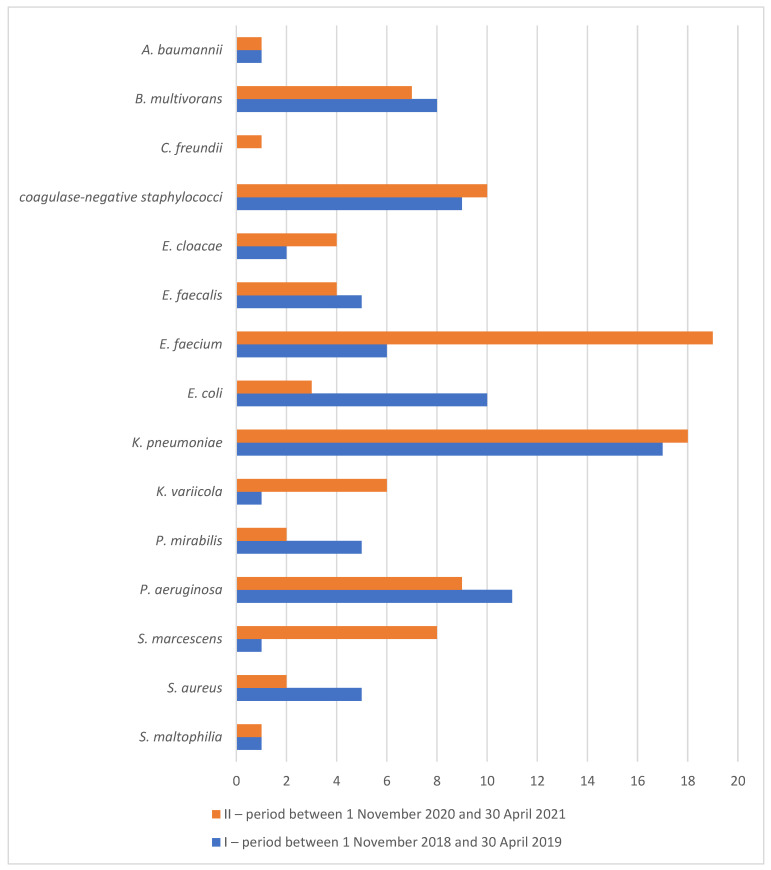
Percentage of species from total number of etiological agents isolated from blood stream, lower respiratory tract and urine in the Department of Anesthesiology, Resuscitation and Intensive Care before (I) and during (II) the COVID-19 pandemic.

**Figure 3 antibiotics-11-00783-f003:**
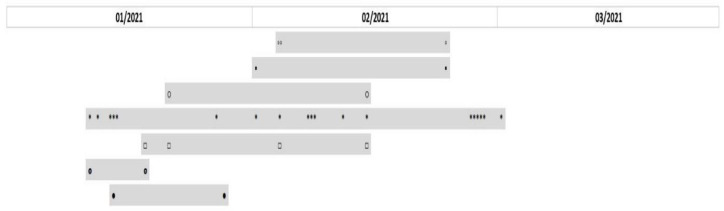
Timeline of the prevalence of *Klebsiella pneumoniae* clones throughout 3 months of the study duration. The grey rectangles represent 7 clonal groups of *Klebsiella pneumoniae* based on their similarity. Isolates are represented by signs ◦▪○*□▯●.

**Figure 4 antibiotics-11-00783-f004:**
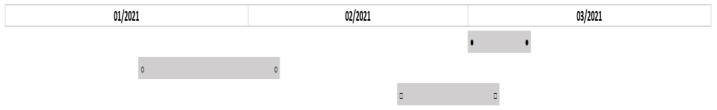
Timeline of the prevalence of *Serratia marcescens* clones throughout the study duration. The grey rectangles represent 3 clonal groups of *Serratia marcescens* based on their similarity. Isolates are represented by signs ●○□.

**Table 1 antibiotics-11-00783-t001:** Characteristics of the examined groups of patients before and during the COVID-19 pandemic. *p*-value in bold type means statistically significant differences between the two groups.

	Patients during the COVID-19 Pandemic (N = 372)	Patients before the COVID-19 Pandemic (N = 189)	*p*-Value
No. of male patients (%)	241 (64)	136 (72)	0.087
Age (median, range IQR)	67 (22–90) (16)	65 (18–97) (20)	0.753
Admissions up to 48 h after admission to hospital (%)	186 (50)	82 (43)	0.138
Admissions from other hospital or department (%)	186 (50)	107 (57)	0.138
No. of patients on mechanical ventilation (%)	249 (67)	184 (97)	<0.0001
No. of patients only on mechanical ventilation (%)	78 (21)	180 (95)	<0.0001
No. of patients on HFNOT (%)	283 (76)	10 (5)	<0.0001
No. of patients only on HFNOT (%)	112 (30)	3 (2)	<0.0001
No. of patient on mechanical ventilation and HFNOT (%)	182 (49)	6 (3)	<0.0001
No. of patients on ECMO (%)	24 (7)	0 (0)	0.004
Mortality on ICU (%)	158 (43)	74 (39)	0.450

**Table 2 antibiotics-11-00783-t002:** The distribution of clinically important bacterial isolates by clinical material in both periods (absolute numbers).

	Material from Lower Respiratory Tract		Material from Upper Respiratory Tract		Blood Cultures and Vascular Catethers		Urine		Material from Wounds, Drains, Punctuations, Pus		Stool		Others	
	I.	II.	I.	II.	I.	II.	I.	II.	I.	II.	I.	II.	I.	II.
*S. maltophilia*	5	12	4	4	0	0	0	0	0	0	0	0	1	0
*S. aureus*	14	16	11	22	2	0	2	0	2	6	0	0	0	1
*S. marcescens*	5	55	2	11	0	8	0	8	0	0	0	0	0	0
*P. aeruginosa*	33	50	16	31	2	4	7	23	8	2	0	2	14	3
*P. mirabilis*	11	8	11	5	0	0	6	6	8	4	0	0	3	1
*K. variicola*	3	40	8	71	0	0	2	10	4	0	0	0	0	2
*K. pneumoniae*	44	126	50	142	5	6	13	30	14	5	0	1	10	7
*E. coli*	23	12	14	18	1	0	11	17	16	3	0	0	3	2
*E. faecium*	8	65	5	22	1	23	13	75	22	5	0	9	5	11
*E. faecalis*	10	19	0	2	1	4	7	11	10	0	0	0	5	2
*E. cloacae*	8	29	14	29	0	3	1	3	3	0	0	0	1	1
coagulase-negative staphylococci	0	0	0	0	30	80	4	7	31	3	0	0	20	14
*C. freundii*	1	3	2	6	0	0	0	2	2	1	0	0	1	0
*B. multivorans*	27	60	26	77	1	1	1	0	4	1	0	0	2	1
*A. baumannii*	4	11	2	29	0	0	1	0	1	0	0	0	0	0
Others	47	36	41	29	6	10	6	8	15	7	3	26	9	1
Total	243	542	206	498	49	139	74	200	140	37	3	38	74	46

Legend: I—period between 1 November 2018 and 30 April 2019, II—period between 1 November 2020 and 30 April 2021.

**Table 3 antibiotics-11-00783-t003:** Bacterial resistance to antibiotics (percentages) before (I) and during (II) the COVID-19 pandemic.

I. Time-Frame 1 November 2018–30 April 2019	I.	II.	*p*	I.	II.	*p*	I.	II.	*p*	I.	II.	*p*	I.	II.	*p*	I.	II.	*p*	I.	II.	*p*
II. Time-Frame 1 November 2020–30 April 2021																					
Species (No. of isolates in I. and II. time-frame)	AMS			CRX			CTX			CTZ			CPM			PPT			MER		
*E. coli* (53; 38)	50	50	1.000	45	39	0.6693	42	39	1.000	42	39	1.000	42	39	1.000	40	32	0.5099	0	0	1.000
*K. pneumoniae* (135; 316)	84	87	0.375	80	81	0.7957	79	79	1.000	79	78	0.804	79	78	0.804	80	81	0.7957	2	0	0.0264
*E. cloacae* (27; 65)	100	100	1.000	100	100	1.000	33	69	0.0002	33	68	0.005	22	42	0.0974	33	68	0.0049	0	0	1.000
*C. freundii* (6; 12)	100	100	1.000	100	100	1.000	50	25	0.3441	50	25	0.3441	0	0	1.000	50	25	0.3441	0	0	1.000
*S. marcescens* (7; 82)	100	100	1.000	100	100	1.000	14	70	0.0066	14	70	0.007	0	70	0.0005	14	68	0.008	0	0	1.000
*P. aeruginosa* (80; 115)	100	100	1.000	100	100	1.000	100	100	1.000	16	21	0.462	19	25	0.3027	28	31	0.634	51	22	0.0001
*B. mutivorans* (59; 139)	100	100	1.000	100	100	1.000	100	100	1.000	5	1	0.080	5	3	0.4273	100	100	1.000	29	2	0.0001
*S. maltophilia* (10; 16)	100	100	1.000	100	100	1.000	100	100	1.000	40	56	0.688	100	100	1.000	100	100	1.000	100	100	1.000
*A. baumannii* (7; 40)	71	0	0.0001	100	100	1.000	100	100	1.000	71	0	0.0001	71	0	0.0001	71	0	0.0001	71	0	0.0001
I. time-frame 1 November 2018–30 April 2019	I.	II.	*p*	I.	II.	*p*	I.	II.	*p*	I.	II.	*p*	I.	II.	*p*	I.	II.	*p*			
II. time-frame 1 November 2020–30 April 2021																					
Species (No. of isolates in I. and II. time-frame)	TIG			COT			GEN			AMI			CIP			COL					
*E. coli* (53; 38)	9	0	0.073	51	29	0.052	19	0	0.004	2	8	0.304	51	39	0.296	0	0	1.000			
*K. pneumoniae* (135; 316)	7	7	1.000	83	87	0.302	75	74	0.907	1	1	1.000	82	76	0.173	4	4	1.000			
*E. cloacae* (27; 65)	7	2	0.205	33	34	1.000	15	29	0.190	0	0	1.000	30	40	0.477	0	3	1.000			
*C. freundii* (6; 12)	0	0	1.000	0	0	1.000	0	17	0.529	0	0	1.000	17	0	0.333	0	0	1.000			
*S. marcescens* (7; 82)	14	100	<0.0001	0	71	0.0004	0	69	0.0005	0	9	1.000	0	67	0.001	100	100	1.000			
*P. aeruginosa* (80; 115)	100	100	1.000	100	100	1.000	23	23	1.000	9	0	0.002	30	26	0.626	3	2	1.000			
*B. mutivorans* (59; 139)	7	4	0.488	0	0	1.000	100	100	1.000	100	100	1.000	100	100	1.000	100	100	1.000			
*S. maltophilia* (10; 16)	0	0	1.000	22	0	0.138	100	100	1.000	100	100	1.000	70	100	0.046	50	88	0.069			
*A. baumannii* (7; 40)	0	0	1.000	71	0	<0.0001	71	0	<0.0001	71	0	<0.0001	71	8	0.001	0	0	1.000			
I. time-frame 1 November 2018–30 April 2019,	I.	II.	*p*	I.	II.	*p*	I.	II.	*p*	I.	II.	*p*	I.	II.	*p*						
II. time-frame 1 November 2020–30 April 2021																					
Species (No. of isolates in I. and II. time-frame)	OXA			COT			ERY			CLI			CIP								

*S. aureus* (31; 45)	0	7	0.266	0	5	0.511	42	24	0.135	42	27	0.216	0	9	0.141						
	GEN			TIG			TEI			VAN			TET								
*S. aureus* (31; 45)	13	7	0.434	0	0	1.000	0	0	1.000	0	0	1.000	3	7	0.641						
I. time-frame 1 November 2018–30 April 2019,	I.	II.	*p*	I.	II.	*p*	I.	II.	*p*	I.	II.	*p*									
II. time-frame 1 November 2020–30 April 2021																					
Species (No. of isolates in I. and II. time-frame)	AMP			TIG			TEI			VAN											
*E. faecalis* (33; 38)	0	0	1.000	0	0	1.000	0	0	1.000	0	0	1.000									
*E. faecium* (54; 210)	100	100	1.000	13	0	<0.0001	19	17	0.841	17	18	1.000									

Legend: I—period between 1 November 2018 and 30 April 2019, II—period between 1 November 2020 and 30 April 2021. AMS—ampicillin/sulbactam, CRX—cefuroxime, CTX—cefotaxime, CTZ—ceftazidime, CPM—cefepime, PPT—piperacillin/tazobactam, MER—meropenem, TIG—tigecycline, COT—co-trimoxazole, GEN—gentamicin, AMI—amikacin, CIP—ciprofloxacin, COL—colistin, OXA—oxacillin, ERY—erythromycin, CLI—clindamycin, TEI—teicoplanin, VAN—vancomycin, AMP—ampicillin.

**Table 4 antibiotics-11-00783-t004:** Consumption of antibacterial agents before (I) and during (II) the COVID-19 pandemic in DDD and as percentage of individual antibiotics in total consumption.

Antibiotics	I.	I.	II.	II.	*p*
	DDD	Percentage	DDD	Percentage	
meropenem	703	19.6	1427	19.3	0.861
piperacillin/tazobactam	301	8.4	570	7.7	0.792
amoxicillin/clavulanic acid	299	8.3	262	3.5	0.020
clarithromycin	294	8.2	641	8.7	0.803
tigecyclin	270	7.5	565	7.6	1.000
gentamicin	250	7	33	0.4	0.243
metronidazol	237	6.6	402	5.4	0.495
sulfamethoxazol/trimethoprim	217	6	370	5	0.708
ampicilin/sulbactam	130	3.6	53	0.7	0.323
ciprofloxacin	124	3.5	150	2	0.705
amikacin	115	3.2	376	5.1	0.618
vancomycin	103	2.9	169	2.3	1.000
ceftazidime	101	2.8	206	2.8	1.000
linezolid	28	0.8	179	2.4	1.000
kolistin	62	1.7	229	3.1	1.000
cefotaxime	60	1.7	1487	20.1	<0.0001
cefuroxime	60	1.7	10	0.1	1.000
other	234	6.5	267	3.8	0.218

Legend: I—period between 1 November 2018 and 30 April 2019, II—period between 1 November 2020 and 30 April 2021.

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
