# Peer review of "Bacterial Resistance to Antibiotics and Clonal Spread in COVID-19-Positive Patients on a Tertiary Hospital Intensive Care Unit, Czech Republic"

_antibiotics, 2022, doi:10.3390/antibiotics11060783_

Round 1
Reviewer 1 Report
The manuscript by Doubravská et al describes the antibiotic resistance and clonal spread of bacterial pathogens during the main waves of the COVID-pandemic in a Department of Anesthesiology, Resuscitation and Intensive Care in comparison to the pre-pandemic period. The authors found clonal spread of bacteria in the intensive care environment. The main concern is the disputable changes in bacterial resistance to antibiotics over the examined periods, most of the observations in the literature indicate an increasing trend in antibiotics resistance during the COVID-19 pandemic. However, as the authors provided possible reasons in the context of discussing their findings, the ambiguous results can nevertheless be of interest. Overall, the manuscript is well written and interesting to read. I have a few minor points:
The title needs to be more specific. Clonal Spread should be indicated in the title (e.g “antibiotic resistance and clonal spread …”
Line 41-43 the sentence should be rephrased as it is not fluid and clear.
Line 50 “5,6” should be “5.6”?
Line 133-136 is a duplication of lines 73-76. I suggest combining the previous sentence as the last paragraph of the introduction.
Line 148 Abbreviation “HAP” needs to be defined in full.
In the last paragraph of the result section “PFGE proved clonal spread of..” Is clonal spread found for all or some of the bacterial strains?
Reviewer 2 Report
1. The study discusses Bacterial resistance to antibiotics in COVID-19-positive patients in the intensive care units during the pre and post-covid-19 periods in the University Hospital Olomouc is one of the largest healthcare facilities in the Czech Republic. This topic is of concern to the public health.
2. However, there are modifications required in the manuscript to improve its readability.
3. Both the title and the abstract need to mention the site/country of the study for readers to understand that this is not the global picture.
4. Line 41: Please modify for clarity of the sentence
5. Line 49: Please provide the current status of the pandemic.
6. Line 67: Remove the word etc.
7. Line 97: replace the word separately with a better contextual word
8. Line 103-130 the content of these paragraphs is focused on the management of COVID-19 rather than the bacterial pathogens and resistance.
9. The details can be reduced and relevant highlights on bacterial resistance in relation to COVID-19 be discussed.
10. Line 142 Please provide evidence-statement that the ethical clearance was waived by which Institution review board?
11. Line 148-151 please spell out HAP on first use?
12. Tables 1 and 2 in the introduction can be removed and summarized in the text of the introduction section, upon citing relevant citations where these were derived,
13. Line 193-207 the PFGE method could be shortened by referring to a standard reference/citation on that method
14. Line 208-210 should provide additional details of the methodology
15, Tables 3 and 4 should be cited in clonological order of mention in the text. Please change the order of tables (Line 226)
16. Figure 1: Add legend and define ARDS
17. Table 5: Please comment on changes in amoxicillin/clavulanic acid
18. Table 5: What is abs. number? How was it measured in the methodology? You may need to use the proper unit for antimicrobial consumption such ass Daily Defined doses
19. The beginning of the discussion section contains text that describes other studies more than the current study. These details can be aligned with the introduction
Please only discuss your results
20. The conclusion should give the condensed message of study findings rather than where the study fails
Instead, a study limitation section can be incorporated into the discussion
21. Also consider changing the decimal place marker from comma (,) to point (.)
Author Response
First, the authors thank the reviewer for the immensely valuable comments, based on which the text has been adapted.
Point 1: Both the title and the abstract need to mention the site/country of the study for readers to understand that this is not the global picture.
Response 1: It has been done according to the reviewer’s comment.
Point 2: Line 41: Please modify for clarity of the sentence.
Response 2: The text has been reformulated. We hope this has improved the coherency of the text.
Point 3: Line 49: Please provide the current status of the pandemic.
Response 3: The data has been updated.
Point 4: Line 67: Remove the word etc.
Response 4: It has been removed.
Point 5: Line 97: Replace the word separately with a better contextual word.
Response 5: It has been done according to the reviewer’s comment.
Point 6: Line 103-130 the content of these paragraphs is focused on the management of COVID-19 rather than the bacterial pathogens and resistance.
Response 6: It has been reduced according to the reviewer’s comment.
Point 7: The details can be reduced and relevant highlights on bacterial resistance in relation to COVID-19 be discussed.
Response 7: In the Introduction section, a paragraph on bacterial infections and antibiotic use in association with COVID-19 has been added, as it could have an impact on increasing bacterial resistance.
Point 8: Line 142 Please provide evidence-statement that the ethical clearance was waived by which Institution review board?
Response 8: We apologize for the mistake and from that arising wrong formulation in the manuscript. Since the study is observational, retrospective and non-interventional and all patients were treated according to standard procedures, there was no informed consent of participants needed. The Ethical Commission gave its approval for the study to be carried out and the sentence regarding this issue has been reformulated in the manuscript. The Ethical Committee approval is available by the principal investigator of the grant Czech Health Research Council project no. NU22-B-112 (funding for this study).
Point 9: Line 148-151 please spell out HAP on first use.
Response 9: It has been done according to the reviewer’s comment.
Point 10: Tables 1 and 2 in the introduction can be removed and summarized in the text of the introduction section, upon citing relevant citations where these were derived
Response 10: Table 1 and 2 were removed and summarized in the text according to the reviewer’s comment.
Point 11: Line 193-207 the PFGE method could be shortened by referring to a standard reference/citation on that method
Response 11: It has been done according to the reviewer’s comment.
Point 12: Line 208-210 should provide additional details of the methodology
Response 12: The details of methodology has been added; we hope this has improved understanding of the methodology.
Point 13: Tables 3 and 4 should be cited in clonological order of mention in the text. Please change the order of tables (Line 226)
Response 13: The tables were renumbered as the first two were deleted. The order of tables was checked to be chronological in the text.
Point 14: Figure 1: Add legend and define ARDS
Response 14: It has been done according to the reviewer’s comment.
Point 15: Table 5: Please comment on changes in amoxicillin/clavulanic acid
Response 15: It has been commented in the Results section according to the reviewer’s comment.
Point 16: Table 5: What is abs. number? How was it measured in the methodology? You may need to use the proper unit for antimicrobial consumption such ass Daily Defined doses
Response 16: The utilization of individual antibiotics was expressed as absolute consumption in defined daily doses (DDD), based on the WHO ATC/DDD classification. In the table 5 the head was changed: instead of abs. number, there is DDD newly.
Point 17: The beginning of the discussion section contains text that describes other studies more than the current study. These details can be aligned with the introduction. Please, only discuss your results
Response 17: It has been rewritten in the Introduction and Discussion section according to the reviewer’s comment.
Point 18: The conclusion should give the condensed message of study findings rather than where the study fails. Instead, a study limitation section can be incorporated into the discussion
Response 18: The strengths and weaknesses of the study are shown at the end of the text in the Discussion section according to the reviewer’s comment. The Conclusion section was slightly reformulated.
Point 19: Also consider changing the decimal place marker from comma (,) to point (.)
Response 19: It has been changed in the whole text according to the reviewer’s comment.
